# Latent Bayesian melding for integrating individual and population models

**Mingjun Zhong, Nigel Goddard, Charles Sutton**
School of Informatics
University of Edinburgh
United Kingdom
{mzhong,nigel.goddard,csutton}@inf.ed.ac.uk

## Abstract

In many statistical problems, a more coarse-grained model may be suitable for population-level behaviour, whereas a more detailed model is appropriate for accurate modelling of individual behaviour. This raises the question of how to integrate both types of models. Methods such as posterior regularization follow the idea of generalized moment matching, in that they allow matching expectations between two models, but sometimes both models are most conveniently expressed as latent variable models. We propose *latent Bayesian melding*, which is motivated by averaging the distributions over populations statistics of both the individual-level and the population-level models under a logarithmic opinion pool framework. In a case study on electricity disaggregation, which is a type of single-channel blind source separation problem, we show that latent Bayesian melding leads to significantly more accurate predictions than an approach based solely on generalized moment matching.

## 1 Introduction

Good statistical models of populations are often very different from good models of individuals. As an illustration, the population distribution over human height might be approximately normal, but to model an individual's height, we might use a more detailed discriminative model based on many features of the individual's genotype. As another example, in social network analysis, simple models like the preferential attachment model [3] replicate aggregate network statistics such as degree distributions, whereas to predict whether two individuals have a link, a social networking web site might well use a classifier with many features of each person's previous history. Of course every model of an individual implies a model of the population, but models whose goal is to model individuals tend to be necessarily more detailed.

These two styles of modelling represent different types of information, so it is natural to want to combine them. A recent line of research in machine learning has explored the idea of incorporating constraints into Bayesian models that are difficult to encode in standard prior distributions. These methods, which include posterior regularization [9], learning with measurements [16], and the generalized expectation criterion [18], tend to follow a moment matching idea, in which expectations of the distribution of one model are encouraged to match values based on prior information.

Interestingly, these ideas have precursors in the statistical literature on simulation models. In particular, Bayesian melding [21] considers applications in which there is a computer simulation $M$ that maps from model parameters $\theta$ to a quantity $\phi = M(\theta)$. For example, $M$ might summarize the output of a deterministic simulation of population dynamics or some other physical phenomenon. Bayesian melding considers the case in which we can build meaningful prior distributions over both $\theta$ and $\phi$. These two prior distributions need to be merged because of the deterministic relationship;

this is done using a logarithmic opinion pool [5]. We show that there is a close connection between Bayesian melding and the later work on posterior regularization, which does not seem to have been recognized in the machine learning literature. We also show that Bayesian melding has the additional advantage that it can be conveniently applied when both individual-level and population-level models contain latent variables, as would commonly be the case, e.g., if they were mixture models or hierarchical Bayesian models. We call this approach *latent Bayesian melding.*

We present a detailed case study of latent Bayesian melding in the domain of energy disaggregation [11, 20], which is a particular type of blind source separation (BSS) problem. The goal of the electricity disaggregation problem is to separate the total electricity usage of a building into a sum of source signals that describe the energy usage of individual appliances. This problem is hard because the source signals are not identifiable, which motivates work that adds additional prior information into the model [14, 15, 20, 25, 26, 8]. We show that the latent Bayesian melding approach allows incorporation of new types of constraints into standard models for this problem, yielding a strong improvement in performance, in some cases amounting to a 50% error reduction over a moment matching approach.

## 2   The Bayesian melding approach

We briefly describe the Bayesian melding approach to integrating prior information in deterministic simulation models [21], which has seen wide application [1, 6, 23]. In the Bayesian modelling context, denote $Y$ as the observation data, and suppose that the model includes unknown variables $S$, which could include model parameters and latent variables. We are then interested in the posterior

$$p(S|Y) = p(Y)^{-1} p(Y|S) p_S(S). \tag{1}$$

However, in some situations, the variables $S$ may be related to a new random variable $\tau$ by a deterministic simulation function $f(\cdot)$ such that $\tau = f(S)$. We call $S$ and $\tau$ input and output variables. For example, in the energy disaggregation problem, the total energy consumption variable $\tau = \sum_{t=1}^{T} S_t^T \mu$ where $S_t$ are the state variables of a hidden Markov model (one-hot encoding) and $\mu$ is a vector containing the mean energy consumption of each state (see Section 5.2). Both $\tau$ and $S$ are random variables, and so in the Bayesian context, the modellers usually choose appropriate priors $p_\tau(\tau)$ and $p_S(S)$ based on prior knowledge. However, given $p_S(S)$, the map $f$ naturally introduces another prior for $\tau$, which is an induced prior denoted by $p_\tau^*(\tau)$. Therefore, there are two different priors for the same variable $\tau$ from different sources, which might not be consistent. In the energy disaggregation example, $p_\tau^*$ is induced by the state variables $S_t$ of the hidden Markov model which is the individual model of a specific household, and $p_\tau$ could be modelled by using population information, e.g. from a national survey — we can think of this as a population model since it combines information from many households. The Bayesian melding approach combines the two priors into one by using the logarithmic pooling method so that the logarithmically pooled prior is $\widetilde{p}_\tau(\tau) \propto p_\tau^*(\tau)^\alpha p_\tau(\tau)^{1-\alpha}$ where $0 \leq \alpha \leq 1$. The prior $\widetilde{p}_\tau$ melds the prior information of both $S$ and $\tau$. In the model (1), the prior $p_S$ does not include information about $\tau$. Thus it is required to derive a melded prior for $S$. If $f$ is invertible, the prior for $S$ can be obtained by using the change-of-variable technique. If $f$ is not invertible, Poole and Raftery [21] heuristically derived a melded prior

$$\widetilde{p}_S(S) = c_\alpha p_S(S) \left( \frac{p_\tau(f(S))}{p_\tau^*(f(S))} \right)^{1-\alpha} \tag{2}$$

where $c_\alpha$ is a constant given $\alpha$ such that $\int \widetilde{p}_S(S) dS = 1$. This gives a new posterior $\widetilde{p}(S|Y) = \widetilde{p}(Y)^{-1} p(Y|S) \widetilde{p}_S(S)$. Note that it is interesting to infer $\alpha$ [22, 7], however we use a fixed value in this paper. So far we have been assuming there are no latent variables in $p_\tau$. We now consider the situation when $\tau$ is generated by some latent variables.

## 3   The latent Bayesian melding approach

It is common that the variable $\tau$ is modelled by a latent variable $\xi$, see the examples in Section 5.2. So we could assume that we have a conditional distribution $p(\tau|\xi)$ and a prior distribution $p_\xi(\xi)$. This defines a marginal distribution $p_\tau(\tau) = \int p_\xi(\xi) p(\tau|\xi) d\xi$. This could be used to produce the

melded prior (2) of the Bayesian melding approach

$$\widetilde{p}_S(S) = c_\alpha p_S(S) \left( \frac{\int p_\tau(f(S)|\xi)p_\xi(\xi)d\xi}{p_\tau^*(f(S))} \right)^{1-\alpha}. \tag{3}$$

The integration in (3) is generally intractable. We could employ the Monte Carlo method to approximate it for a fixed $\tau$. However, importantly we are also interested in inferring the latent variable $\xi$ which is meaningful for example in the energy disaggregation problem. When we are interested in finding the maximum a posteriori (MAP) value of the posterior where $\widetilde{p}_S(S)$ was used as the prior, we propose to use a rough approximation $\int p_\xi(\xi)p_\tau(\tau|\xi)d\xi \approx \max_\xi p_\xi(\xi)p_\tau(\tau|\xi)$. This leads to an approximate prior

$$\widetilde{p}_S(S) \approx \max_\xi \widetilde{p}_{S,\xi}(S,\xi) = \max_\xi c_\alpha p_S(S) \left( \frac{p_\tau(f(S)|\xi)p_\xi(\xi)}{p_\tau^*(f(S))} \right)^{1-\alpha}. \tag{4}$$

To obtain this approximate prior for $S$, the joint prior $\widetilde{p}_{S,\xi}(S,\xi)$ has to exist, and so we show that it does exist under certain conditions by the following theorem. We assume that $S$ and $\xi$ are continuous random variables, and that both $p_\tau^*$ and $p_\tau$ are positive and share the same support. Also, $E_{p_S(S)}[\cdot]$ denotes the expectation with respect to $p_S$.

**Theorem 1.** *If* $E_{p_S(S)}\left[ \frac{p_\tau(f(S))}{p_\tau^*(f(S))} \right] < \infty$, *then a constant* $c_\alpha < \infty$ *exists such that* $\int \widetilde{p}_{S,\xi}(S,\xi)d\xi dS = 1$, *for any fixed* $\alpha \in [0,1]$.

The proof can be found in the supplementary materials. In (4) we heuristically derived an approximate joint prior $\widetilde{p}_{S,\xi}$. Interestingly, if $\xi$ and $S$ are independent conditional on $\tau$, we can show as follows that $\widetilde{p}_{S,\xi}$ is a limit distribution derived from a joint distribution of $\xi$ and $S$ induced by $\tau$. To see this, we derive a joint prior for $S$ and $\xi$,

$$
\begin{aligned}
p_{S,\xi}(S,\xi) &= \int p(S,\xi|\tau)p_\tau(\tau)d\tau = \int p(S|\tau)p(\xi|\tau)p_\tau(\tau)d\tau \\
&= \int \frac{p(\tau|S)p_S(S)}{p_\tau^*(\tau)} \frac{p(\tau|\xi)p_\xi(\xi)}{p_\tau(\tau)} p_\tau(\tau)d\tau = p_S(S)p_\xi(\xi) \int p(\tau|S) \frac{p(\tau|\xi)}{p_\tau^*(\tau)} d\tau.
\end{aligned}
$$

For a deterministic simulation $\tau = f(S)$, the distribution $p(\tau|S) = p(\tau|S, \tau = f(S))$ is ill-defined due to the Borel's paradox [24]. The distribution $p(\tau|S)$ depends on the parameterization. We assume that $\tau$ is uniform on $[f(S) - \delta, f(S) + \delta]$ conditional on $S$ and $\delta > 0$, and the distribution is then denoted by $p_\delta(\tau|S)$. The marginal distribution is $p_\delta(\tau) = \int p_\delta(\tau|S)p_S(S)dS$. Denote $g(\tau) = \frac{p(\tau|\xi)}{p_\tau^*(\tau)}$ and $g_\delta(\tau) = \frac{p(\tau|\xi)}{p_\delta(\tau)}$. Then we have the following theorem.

**Theorem 2.** *If* $\lim_{\delta \to 0} p_\delta(\tau) = p_\tau^*(\tau)$, *and* $g_\delta(\tau)$ *has bounded derivatives in any order, then* $\lim_{\delta \to 0} \int p_\delta(\tau|S)g_\delta(\tau)d\tau = g(f(S))$.

See the supplementary materials for the proof. Under this parameterization, we denote $\hat{p}_{S,\xi}(S,\xi) = p_S(S)p_\xi(\xi)\lim_{\delta \to 0}\int p_\delta(\tau|S)g_\delta(\tau)d\tau = p_S(S)p_\xi(\xi)\frac{p(f(S)|\xi)}{p_\tau^*(f(S))}$. By applying the logarithmic pooling method, we have a joint prior

$$\widetilde{p}_{S,\xi}(S,\xi) = c_\alpha \left( p_S(S) \right)^\alpha \left( \hat{p}_{S,\xi}(S,\xi) \right)^{1-\alpha} = c_\alpha p_S(S) \left( \frac{p_\tau(f(S)|\xi)p_\xi(\xi)}{p_\tau^*(f(S))} \right)^{1-\alpha}.$$

Since the joint prior blends the variable $S$ and the latent variable $\xi$, we call this approximation the latent Bayesian melding (LBM) approach, which gives the posterior $\widetilde{p}(S,\xi|Y) = \widetilde{p}(Y)^{-1}p(Y|S)\widetilde{p}_{S,\xi}(S,\xi)$. Note that if there are no latent variables, then latent Bayesian melding collapses to the Bayesian melding approach. In section 6 we will apply this method to an energy disaggregation problem for integrating population information with an individual model.

## 4 Related methods

We now discuss possible connections between Bayesian melding (BM) and other related methods. Recently in machine learning, moment matching methods have been proposed, e.g., posterior regularization (PR) [9], learning with measurements [16] and the generalized expectation criterion [18].

These methods share the common idea that the Bayesian models (or posterior distributions) are constrained by some observations or measurements to obtain a least-biased distribution. The idea is that the system we are modelling is too complex and unobservable, and thus we have limited prior information. To alleviate this problem, we assume we can obtain some observations of the system in some way, e.g., by experiments, for example those observations could be the mean values of the functions of the variables. Those observations could then guide the modelling of the system. Interestingly, a very similar idea has been employed in the bias correction method in information theory and statistics [12, 10, 19], where the least-biased distribution is obtained by optimizing the Kullback-Leibler divergence subject to the moment constraints. Note that the bias correction method in [17] is different to others where the bias of a consistent estimator was corrected when the bias function could be estimated.

We now consider the posteriors derived by PR and BM. In general, given a function $f(S)$ and values $b_i$, PR solves the constrained problem

$$\underset{\widetilde{p}}{\text{minimize}} \quad KL(\widetilde{p}(S)||p(S|Y)) \quad \text{subject to} \quad E_{\widetilde{p}}(m_i(f(S))) - b_i \leq \delta_i, ||\delta_i|| \leq \epsilon; i = 1, 2, \cdots, I.$$

where $m_i$ could be any function such as a power function. This gives an optimal posterior $\widetilde{p}_{PR}(S) = Z(\lambda)^{-1} p(Y|S) p(S) \prod_{i=1}^{I} \exp(-\lambda_i m_i(f(S)))$ where $Z(\lambda)$ is the normalizing constant. BM has a deterministic simulation $f(S) = \tau$ where $\tau \sim p_\tau$. The posterior is then $\widetilde{p}_{BM}(S) = Z(\alpha)^{-1} p(Y|S) p(S) \left( \frac{p_\tau(f(S))}{p_\tau^*(f(S))} \right)^{1-\alpha}$. They have a similar form and the key difference is the last factor which is derived from the constraints or the deterministic simulation. $\widetilde{p}_{PR}$ and $\widetilde{p}_{BM}$ are identical, if $-\sum_{i=1}^{I} \lambda_i m_i(f(S)) = (1-\alpha) \log \frac{p_\tau(f(S))}{p_\tau^*(f(S))}$.

The difference between BM and LBM is the latent variable $\xi$. We could perform BM by integrating out $\xi$ in (3), but this is computationally expensive. Instead, LBM jointly models $S$ and $\xi$ allowing possibly joint inference, which is an advantage over BM.

# 5 The energy disaggregation problem

In energy disaggregation, we are given a time series of energy consumption readings from a sensor. We consider the energy measured in watt hours as read from a household's electricity meter, which is denoted by $Y = (Y_1, Y_2, \cdots, Y_T)$ where $Y_t \in R_+$. The recorded energy signal $Y$ is assumed to be the aggregation of the consumption of individual appliances in the household. Suppose there are $I$ appliances, and the energy consumption of each appliance is denoted by $X_i = (X_{i1}, X_{i2}, \cdots, X_{iT})$ where $X_{it} \in R_+$. The observed aggregate signal is assumed to be the sum of the component signals so that $Y_t = \sum_{i=1}^{I} X_{it} + \epsilon_t$ where $\epsilon_t \sim \mathcal{N}(0, \sigma^2)$. Given $Y$, the task is to infer the unknown component signals $X_i$. This is essentially the single-channel BSS problem, for which there is no unique solution. It can also be useful to add an extra component $U = (U_1, U_2, \cdots, U_T)$ to model the unknown appliances to make the model more robust as proposed in [15]. The prior of $U_t$ is defined as $p(U) = \frac{1}{v^{2(T-1)}} \exp \left\{ -\frac{1}{2v^2} \sum_{t=1}^{T-1} |U_{t+1} - U_t| \right\}$. The model then has a new form $Y_t = \sum_{i=1}^{I} X_{it} + U_t + \epsilon_t$. A natural way to represent this model is as an additive factorial hidden Markov model (AFHMM) where the appliances are treated as HMMs [15, 20, 26]; this is now described.

## 5.1 The additive factorial hidden Markov model

In the AFHMM, each component signal $X_i$ is represented by a HMM. We suppose there are $K_i$ states for each $X_{it}$, and so the state variable is denoted by $Z_{it} \in \{1, 2, \cdots, K_i\}$. Since $X_i$ is a HMM, the initial probabilities are $\pi_{ik} = P(Z_{i1} = k)$ $(k = 1, 2, \cdots, K_i)$ where $\sum_{k=1}^{K_i} \pi_{ik} = 1$; the mean values are $\mu_i = \{\mu_1, \mu_2, \cdots, \mu_{K_i}\}$ such that $X_{it} \in \mu_i$; the transition probabilities are $P^{(i)} = (p_{jk}^{(i)})$ where $p_{jk}^{(i)} = P(Z_{it} = j | Z_{i,t-1} = k)$ and $\sum_{j=1}^{K_i} p_{jk}^{(i)} = 1$. We denote all these parameters $\{\pi_i, \mu_i, P^{(i)}\}$ by $\theta$. We assume they are known and can be learned from the training data. Instead of using $Z$, we could use a binary vector $S_{it} = (S_{it1}, S_{it2}, \cdots, S_{itK_i})^T$ to represent the variable $Z$ such that $S_{itk} = 1$ when $Z_{it} = k$ and for all $S_{itj} = 0$ when $j \neq k$. Then we are interested in inferring the states $S_{it}$ instead of inferring $X_{it}$ directly, since $X_{it} = S_{it}^T \mu_i$. Therefore

we want to make inference over the posterior distribution

$$P(S, U, \sigma^2 | Y, \theta) \quad \propto \quad p(Y|S, U, \sigma^2) P(S|\theta) p(U) p(\sigma^2)$$

where the HMM defines the prior of the states $P(S|\theta) \quad \propto \quad \prod_{i=1}^{I} \prod_{k=1}^{K_i} \pi_{ik}^{S_{i1k}} \times$ $\prod_{t=2}^{T} \prod_{i=1}^{I} \prod_{k,j} \left( p_{kj}^{(i)} \right)^{S_{itk} S_{i,t-1,j}}$, the inverse noise variance is assumed to be a Gamma distribution $p(\sigma^{-2}) \propto (\sigma^{-2})^{\alpha-1} \exp\left\{ -\beta \sigma^{-2} \right\}$, and the data likelihood has the Gaussian form $p(Y|S, U, \sigma^2, \theta) = |2\pi\sigma^2|^{-\frac{T}{2}} \exp\left\{ -\frac{1}{2\sigma^2} \sum_{t=1}^{T} \left( Y_t - \sum_{i=1}^{I} S_{it}^T \mu_i - U_t \right)^2 \right\}$. To make the MAP inference over $S$, we relax the binary variable $S_{itk}$ to be continuous in the range $[0, 1]$ as in [15, 26]. It has been shown that incorporating domain knowledge into AFHMM can help to reduce the identifiability problem [15, 20, 26]. The domain knowledge we will incorporate using LBM is the summary statistics.

## 5.2 Population modelling of summary statistics

In energy disaggregation, it is useful to provide a summaries of energy consumption to the users. For example, it would be useful to show the householders the total energy they had consumed in one day for their appliances, the duration that each appliance was in use, and the number of times that they had used these appliances. Since there already exists data about typical usage of different appliances [4], we can employ these data to model the distributions of those summary statistics.

We denote those desired statistics by $\tau = \{\tau_i\}_{i=1}^{I}$, where $i$ denotes the appliances. For appliance $i$, we assume we have measured some time series from different houses for many days. This is always possible because we can collect them from public data sets, e.g., the data reviewed in [4]. We can then empirically obtain the distributions of those statistics. The distribution is represented by $p_m(\tau_{im}|\Gamma_{im}, \eta_{im})$ where $\Gamma_{im}$ represents the empirical quantities of the statistic $m$ of the appliance $i$ which can be obtained from data and $\eta_{im}$ are the latent variables which might not be known. Since $\eta_{im}$ are variables, we can employ a prior distribution $p(\eta_{im})$.

We now give some examples of those statistics. **Total energy consumption:** The total energy consumption of an appliance can be represented as a function of the states of HMM such that $\tau_i = \sum_{t=1}^{T} S_{it}^T \mu_i$. **Duration of appliance usage:** The duration of using the appliance $i$ can also be represented as a function of states $\tau_i = \Delta t \sum_{t=1}^{T} \sum_{k=2}^{K_i} S_{itk}$ where $\Delta t$ represents the sampling duration for a data point of the appliances, and we assume that $S_{it1}$ represents the off state which means the appliance was turned off. **Number of cycles:** The number of cycles (the number of times an appliance is used) can be counted by computing the number of alterations from OFF state to ON such that $\tau_i = \sum_{t=2}^{T} \sum_{k=2}^{K_i} I(S_{itk} = 1, S_{i,t-1,1} = 0)$.

Let the binary vector $\xi_i = (\xi_{i1}, \xi_{i2}, \cdots, \xi_{ic}, \cdots, \xi_{iC_i})$ represent the number of cycles, where $\xi_{ic} = 1$ means that the appliance $i$ had been used $c$ cycles, and $\sum_{c=1}^{C_i} \xi_{ic} = 1$. (Note $\xi_i$ is an example of $\eta_i$ in this case.) To model these statistics in our LBM framework, the latent variable that we use is the number of cycles $\xi$. The distributions of $\tau_i$ could be empirically modelled by using the observation data. One approach is to assume a Gaussian mixture density such that $p(\tau_i|\xi_i) = \sum_{c=1}^{C_i} p(\xi_{ic} = 1) p_c(\tau_i|\Gamma_i)$, where $\sum_{c=1}^{C_i} p(\xi_{ic} = 1) = 1$ and $p_c$ is the Gaussian component density. Using the mixture Gaussian, we basically assume that, for an appliance, given the number of cycles the total energy consumption is modelled by a Gaussian with mean $\overline{\mu}_{ic}$ and variance $\overline{\sigma}_{ic}^2$. A simpler model would be a linear regression model such that $\tau_i = \sum_{c=1}^{C_i} \xi_{ic} \overline{\mu}_{ic} + \epsilon_i$ where $\epsilon_i \sim \mathcal{N}(0, \sigma_i^2)$. This model assumes that given the number of cycles the total energy consumption is close to the mean $\overline{\mu}_{ic}$. The mixture model is more appropriate than the regression model, but the inference is more difficult.

When $\tau_i$ represents the number of cycles for appliance $i$, we can use $\tau_i = \sum_{c=1}^{C_i} \overline{c}_{ic} \xi_{ic}$ where $\overline{c}_{ic}$ represents the number of cycles. When the state variables $S_i$ are relaxed to $[0, 1]$, we can then employ a noise model such that $\tau_i = \sum_{c=1}^{C_i} \overline{c}_{ic} \xi_{ic} + \epsilon_i$ where $\epsilon \sim \mathcal{N}(0, \sigma_i^2)$. We model $\xi_i$ with a discrete distribution such that $P(\xi_i) = \prod_{c=1}^{C_i} p_{ic}^{\xi_{ic}}$ where $p_{ic}$ represents the prior probability of the number of cycles for the appliance $i$, which can be obtained from the training data. We now show that how to use the LBM to integrate the AFHMM with these population distributions.

# 6 The latent Bayesian melding approach to energy disaggregation

We have shown that the summary statistics $\tau$ can be represented as a deterministic function of the state variable of HMMs $S$ such that $\tau = f(S)$, which means that the $\tau$ itself can be represented as a latent variable model. We could then straightforwardly employ the LBM to produce a joint prior over $S$ and $\xi$ such that $\widetilde{p}_{S,\xi}(S,\xi) = c_\alpha p_S(S) \left( \frac{p_\tau(f(S)|\xi)p(\xi)}{p_\tau^*(f(S))} \right)^{1-\alpha}$. Since in our model $f$ is not invertible, we need to generate a proper density for $p_\tau^*$. One possible way is to generate $N$ random samples $\{S^{(n)}\}_{n=1}^N$ from the prior $p_S(S)$ which is a HMM, and then $p_\tau^*$ can be modelled by using kernel density estimation. However, this will make the inference difficult. Instead, we employ a Gaussian density $p_{\tau_{im}}^*(\tau_{im}) = \mathcal{N}(\hat{\mu}_{im}, \hat{\sigma}_{im}^2)$ where $\hat{\mu}_{im}$ and $\hat{\sigma}_{im}^2$ are computed from $\{S^{(n)}\}_{n=1}^N$. The new posterior distribution of LBM thus has the form

$$
\begin{aligned}
p(S, U, \Sigma | Y, \theta) &\propto p(\Sigma)p(U)\widetilde{p}_{S,\xi}(S,\xi)p(Y|S,U,\sigma^2) \\
&= p(\Sigma)p(U)c_\alpha p_S(S) \left( \frac{p_\tau(f(S)|\xi)p(\xi)}{p_\tau^*(f(S))} \right)^{1-\alpha} p(Y|S,U,\sigma^2)
\end{aligned}
$$

where $\Sigma$ represents the collection of all the noise variances. All the inverse noise variances employ the Gamma distribution as the prior. We are interested in inferring the MAP values. Since the variables $S$ and $\xi$ are binary, we have to solve a combinatorial optimization problem which is intractable, so we solve a relaxed problem as in [15, 26]. Since $\log p_S(S)$ is not convex, we employ the relaxation method of [15]. So a new $K_i \times K_i$ variable matrix $H^{it} = (h_{jk}^{it})$ is introduced such that $h_{jk}^{it} = 1$ when $S_{i,t-1,k} = 1$ and $S_{itj} = 1$ and otherwise $h_{jk}^{it} = 0$. Under these constraints, we then obtain $\log p_S(S) = \log p(S,H) = \sum_{i=1}^I S_{i1}^T \log \pi_i + \sum_{i,t,k,j} h_{jk}^{it} \log p_{jk}^{(i)}$; this is now linear. We optimize the log-posterior which is denoted by $\mathcal{L}(S, H, U, \Sigma, \xi)$. The constraints for those variables are represented as sets $\mathcal{Q}_S = \left\{ \sum_{k=1}^{K_i} S_{itk} = 1, S_{itk} \in [0,1], \forall i, t \right\}$, $\mathcal{Q}_\xi = \left\{ \sum_{c=1}^{C_i} \xi_{ic} = 1, \xi_{ic} \in [0,1], \forall i \right\}$, $\mathcal{Q}_{H,S} = \left\{ \sum_{l=1}^{K_i} H_{l.}^{it} = S_{i,t-1}^T, \sum_{l=1}^{K_i} H_{.l}^{it} = S_{it}, h_{jk}^{it} \in [0,1], \forall i, t \right\}$, and $\mathcal{Q}_{U,\Sigma} = \left\{ U \geq 0, \Sigma \geq 0, \sigma_{im}^2 < \hat{\sigma}_{im}^2, \forall i, m \right\}$. Denote $\mathcal{Q} = \mathcal{Q}_S \cup \mathcal{Q}_\xi \cup \mathcal{Q}_{H,S} \cup \mathcal{Q}_{U,\Sigma}$. The relaxed optimization problem is then

$$
\underset{S,H,U,\Sigma,\xi}{\text{maximize}} \quad \mathcal{L}(S, H, U, \Sigma, \xi) \quad \text{subject to} \quad \mathcal{Q}.
$$

We oberved that every term in $\mathcal{L}$ is either quadratic or linear when $\Sigma$ are fixed, and the solutions for $\Sigma$ are deterministic when the other variables are fixed. The constraints are all linear. Therefore, we optimize $\Sigma$ while fixing all the other variables, and then optimize all the other variables simultaneously while fixing $\Sigma$. This optimization problem is then a convex quadratic program (CQP), for which we use MOSEK [2]. We denote this method by AFHMM+LBM.

# 7 Experimental results

We have incorporated population information into the AFHMM by employing the latent Bayesian melding approach. In this section, we apply the proposed model to the disaggregation problem. We will compare the new approach with the AFHMM+PR [26] using the set of statistics $\tau$ described in Section 5.2. The key difference between our method AFHMM+LBM and AFHMM+PR is that AFHMM+LBM models the statistics $\tau$ conditional on the number of cycles $\xi$.

## 7.1 The HES data

We apply AFHMM, AFHMM+PR and AFHMM+LBM to the Household Electricity Survey (HES) data[1]. This data set was gathered in a recent study commissioned by the UK Department of Food and Rural Affairs. The study monitored 251 households, selected to be representative of the population, across England from May 2010 to July 2011 [27]. Individual appliances were monitored, and in some households the overall electricity consumption was also monitored. The data were monitored

Table 1: Normalized disaggregation error (NDE), signal aggregate error (SAE), duration aggregate error (DAE), and cycle aggregate error (CAE) by AFHMM+PR and AFHMM+LBM on synthetic mains in HES data.

| Methods | NDE | SAE | DAE | CAE | Time (s) |
|---------|-----|-----|-----|-----|----------|
| AFHMM | 1.45± 0.88 | 1.42± 0.39 | 1.56±0.23 | 1.41±0.31 | 179.3±1.9 |
| AFHMM+PR | 0.87± 0.21 | 0.86± 0.39 | 0.83±0.53 | 1.57±0.66 | 195.4±3.2 |
| AFHMM+LBM | 0.89± 0.49 | 0.87± 0.37 | 0.76±0.32 | 0.79±0.35 | 198.1±3.1 |

Table 2: Normalized disaggregation error (NDE), signal aggregate error (SAE), duration aggregate error (DAE), and cycle aggregate error (CAE) by AFHMM+PR and AFHMM+LBM on mains in HES data.

| Methods | NDE | SAE | DAE | CAE | Time (s) |
|---------|-----|-----|-----|-----|----------|
| AFHMM | 1.90±1.16 | 2.26±0.86 | 1.91±0.67 | 1.12 ±0.17 | 170.8±33.3 |
| AFHMM+PR | 0.91±0.11 | 0.67± 0.07 | 0.68± 0.18 | 1.65 ±0.49 | 214.2±38.1 |
| AFHMM+LBM | 0.77±0.23 | 0.68± 0.19 | 0.61± 0.22 | 0.98±0.32 | 224.8±34.8 |

every 2 or 10 minutes for different houses. We used only the 2-minute data. We then used the individual appliances to train the model parameters $\theta$ of the AFHMM, which will be used as the input to the models for disaggregation. Note that we assumed the HMMs have 3 states for all the appliances. This number of states is widely applied in energy disaggregation problems, though our method could easily be applied to larger state spaces. In the HES data, in some houses the overall electricity consumption (the mains) was monitored. However, in most houses, only a subset of individual appliances were monitored, and the total electricity readings were not recorded.

**Generating the population information**: Most of the houses in HES did not monitor the mains readings. They all recorded the individual appliances consumption. We used a subset of the houses to generate the population information of the individual appliances. We used the population information of total energy consumption, duration of appliance usage and the number of cycles in a time period. In our experiments, the time period was one day. We modelled the distributions of these summary statistics by using the methods described in the Section 5.2, where the distributions were Gaussian. All the required quantities for modelling these distributions were generated by using the samples of the individual appliances.

**Houses without mains readings**: In this experiment, we randomly selected one hundred households, and one day's usage was used as test data for each household. Since no mains readings were monitored in these houses, we added up the appliance readings to generate synthetic mains readings. We then applied the AFHMM, AFHMM+PR and AFHMM+LBM to these synthetic mains to predict the individual appliance usage. To compare these three methods, we employed four error measures. Denote $\hat{x}_i$ as the inferred signal for the appliance usage $x_i$. One measure is the normalized disaggregation error (NDE): $\frac{\sum_{it}(x_{it}-\hat{x}_{it})^2}{\sum_{it}x_{it}^2}$. This measures how well the method predicts the energy consumption at every time point. However, the householders might be more interested in the summaries of the appliance usage. For example, in a particular time period, e.g, one day, people are interested in the total energy consumption of the appliances, the total time they have been using those appliances and how many times they have used them. We thus employ $\frac{1}{I}\sum_{i=1}^{I}\frac{|\hat{r}_i-r_i|}{\sum_i r_i}$ as the signal aggregate error (SAE), the duration aggregate error (DAE) or the cycle aggregate error (CAE), where $r_i$ represents the total energy consumption, the duration or the number of cycles, respectively, and $\hat{r}_i$ represents the predicted summary statistics.

All the methods were applied to the synthetic data. Table 1 shows the overall error computed by these methods. We see that both the methods using prior information improved over the base line method AFHMM. The AFHMM+PR and AFHMM+LBM performed similarly in terms of NDE and SAE, but AFHMM+LBM improved over AFHMM+PR in terms of DAE (8%) and CAE (50%).

**Houses with mains readings**: We also applied those methods to 6 houses which have mains readings. We used 10 days data for each house, and the recorded mains readings were used as the input to the models. All the methods were used to predict the appliance consumption. Table 2 shows the

Table 3: Normalized disaggregation error (NDE), signal aggregate error (SAE), duration aggregate error (DAE), and cycle aggregate error (CAE) by AFHMM+PR and AFHMM+LBM on UK-DALE data.

| METHODS | NDE | SAE | DAE | CAE | TIME (S) |
|---|---|---|---|---|---|
| AFHMM | 1.57±1.16 | 1.99±0.52 | 2.81±0.79 | 1.37 ± 0.28 | 118.6±23.1 |
| AFHMM+PR | 0.83±0.27 | 0.82± 0.38 | 1.68± 1.21 | 1.90 ±0.52 | 120.4±25.3 |
| AFHMM+LBM | 0.84±0.25 | 0.89± 0.38 | 0.49± 0.33 | 0.59±0.21 | 123.1±25.8 |

error of each house and also the overall errors. This experiment is more realistic than the synthetic mains readings, since the real mains readings were used as the input. We see that both the methods incorporating prior information have improved over the AFHMM in terms of NDE, SAE and DAE. The AFHMM+PR and AFHMM+LBM have the similar results for SAE. AFHMM+LBM is improved over AFHMM+PR for NDE (15%), DAE (10%) and CAE (40%).

## 7.2 UK-DALE data

In the previous section we have trained the model using the HES data, and applied the models to different houses of the same data set. A more realistic situation is to train the model in one data set, and apply the model to a different data set, because it is unrealistic to expect to obtain appliance-level data from every household on which the system will be deployed. In this section, we use the HES data to train the model parameters of the AFHMM, and model the distribution of the summary statistics. We then apply the models to the UK-DALE dataset [13], which was also gathered from UK households, to make the predictions. There are five houses in UK-DALE, and all of them have mains readings and as well as the individual appliance readings. All the mains meters were sampled every 6 seconds and some of them also sampled at a higher rate, details of the data and how to access it can be found in [13]. We employ three of the houses for analysis in our experiments (houses 1, 2 & 5 in the data). The other two houses were excluded because the correlation between the sum of submeters and mains is very low, which suggests that there might be recording errors in the meters. We selected 7 appliances for disaggregation, based on those that typically use the most energy. Since the sample rate of the submeters in the HES data is 2 minutes, we downsampled the signal from 6 seconds to 2 minutes for the UK-DALE data. For each house, we randomly selected a month for analysis. All the four methods were applied to the mains readings. For comparison purposes, we computed the NDE, SAE, DAE and CAE errors of all three methods, averaged over 30 days. Table 3 shows the results. The results are consistent with the results of the HES data. Both the AFHMM+PR and AFHMM+LBM improve over the basic AFHMM, except that AFHMM+PR did not improve the CAE. As for HES testing data, AFHMM+PR and AFHMM+LBM have similar results on NDE and SAE. And AFHMM+LBM again improved over AFHMM+PR in DAE (70%) and CAE (68%). These results are consistent in suggesting that incorporating population information into the model can help to reduce the identifiability problem in single-channel BSS problems.

## 8 Conclusions

We have proposed a latent Bayesian melding approach for incorporating population information with latent variables into individual models, and have applied the approach to energy disaggregation problems. The new approach has been evaluated by applying it to two real-world electricity data sets. The latent Bayesian melding approach has been compared to the posterior regularization approach (a case of the Bayesian melding approach) and AFHMM. Both the LBM and PR have significantly lower error than the base line method. LBM improves over PR in predicting the duration and the number of cycles. Both methods were similar in NDE and the SAE errors.

**Acknowledgments**

This work is supported by the Engineering and Physical Sciences Research Council, UK (grant numbers EP/K002732/1 and EP/M008223/1).

## Footnotes

[1]The HES dataset and information on how the raw data was cleaned can be found from https://www.gov.uk/government/publications/household-electricity-survey.

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
