[Supplementary Material]

# Supplementary materials for latent Bayesian melding for integrating individual and population models

**Mingjun Zhong, Nigel Goddard, Charles Sutton**
School of Informatics
University of Edinburgh
United Kingdom
{mzhong,nigel.goddard,csutton}@inf.ed.ac.uk

**Theorem 1.** *If* $E_{p_S(S)}\left[\frac{p_\tau(f(S))}{p_\tau^*(f(S))}\right] < \infty$, *then a constant* $c_\alpha < \infty$ *exists such that* $\int \widetilde{p}_{S,\xi}(S,\xi)d\xi dS = 1$, *for any fixed* $\alpha \in [0,1]$.

*Proof.* If $\alpha = 1$, then $c_\alpha = 1$. If $\alpha = 0$, then,

$$\int p_S(S)\left(\frac{p_\tau(f(S)|\xi)p(\xi)}{p_\tau^*(f(S))}\right)d\xi dS = \int p_S(S)\frac{p_\tau(f(S))}{p_\tau^*(f(S))}\frac{p_\tau(f(S)|\xi)p(\xi)}{p_\tau(f(S))}d\xi dS$$

$$= \int p_S(S)\frac{p_\tau(f(S))}{p_\tau^*(f(S))}dS < \infty$$

Now we look at $\alpha \in (0,1)$. Firstly, for $x > 0$, if $\alpha \in (0,1)$, then $g(x) = x^{1-\alpha}$ is a concave function, because $g(x)'' = -\alpha(1-\alpha)x^{-\alpha-1} < 0$. Similarly, $x^\alpha$ is also a concave function. Then we have

$$\int p_S(S)\left(\frac{p_\tau(f(S)|\xi)p(\xi)}{p_\tau^*(f(S))}\right)^{1-\alpha}d\xi dS$$

$$= \int p_S(S)\left(\frac{p_\tau(f(S))}{p_\tau^*(f(S))}\right)^{1-\alpha}E_{p(\xi|f(S))}\left[\left(\frac{p_\tau(f(S))}{p_\tau(f(S)|\xi)p(\xi)}\right)^\alpha\right]dS$$

$$\leq \int p_S(S)\left(\frac{p_\tau(f(S))}{p_\tau^*(f(S))}\right)^{1-\alpha}\left[E_{p(\xi|f(S))}\left(\frac{p_\tau(f(S))}{p_\tau(f(S)|\xi)p(\xi)}\right)\right]^\alpha dS$$

$$= \int p_S(S)\left(\frac{p_\tau(f(S))}{p_\tau^*(f(S))}\right)^{1-\alpha}dS$$

$$\leq \left[E_{p_S(S)}\left(\frac{p_\tau(f(S))}{p_\tau^*(f(S))}\right)\right]^{1-\alpha}$$

$$< \infty$$

where Jensen's inequality has been applied twice. Therefore, $c_\alpha < \infty$ exists, satisfying $\int \widetilde{p}_{\xi,S}(\xi,S)d\xi dS = 1$. □

**Theorem 2.** *If* $\lim_{\delta \to 0} p_\delta(\tau) = p_\tau^*(\tau)$, *and* $g_\delta(\tau)$ *has bounded derivatives in any order, then* $\lim_{\delta \to 0} \int p_\delta(\tau|S) g_\delta(\tau) d\tau = g(f(S))$.

*Proof.* Since $\tau$ is an Uniform distribution on $[f(S) - \delta, f(S) + \delta]$ conditional on $S$ and $\delta$, we could draw $N$ samples for $\tau$ such that $\tau_i = f(S) + (2u_i - 1)\delta$ where $u_i$ is a sample drawn from the standard Uniform distribution, where $i = 1, 2, \cdots, N$. By using Monte Carlo approximation and Taylor's expansion, we have

$$\lim_{\delta \to 0} \int p_\delta(\tau|S) g_\delta(\tau) d\tau$$

$$= \lim_{\delta \to 0} \int p(\tau \in (f(S) - \delta, f(S) + \delta)) g_\delta(\tau) d\tau$$

$$= \lim_{\delta \to 0} \lim_{N \to \infty} \frac{1}{N} \sum_{i=1}^{N} g_\delta(f(S) + (2u_i - 1)\delta)$$

$$= \lim_{\delta \to 0} \lim_{N \to \infty} \left\{ g_\delta(f(S)) + g_\delta'(f(S))\delta \frac{1}{N} \sum_{i=1}^{N} (2u_i - 1) + \frac{1}{2!} g_\delta''(f(S))\delta^2 \frac{1}{N} \sum_{i=1}^{N} (2u_i - 1)^2 + \cdots \right\}$$

$$= \lim_{\delta \to 0} g_\delta(f(S))$$

$$= g(f(S)).$$

This holds, since $|2u_i - 1|^k \leq 1$ $(k = 1, 2, 3, \cdots)$ and $\lim_{N \to \infty} \sum_{i=1}^{N} \frac{1}{N} |2u_i - 1|^k \leq 1$, $\sum_{i=1}^{N} \frac{1}{N} (2u_i - 1)^k$ converges absolutely when $N \to \infty$. $\qquad \square$