[Reviews · NeurIPS 2015]

Submitted by Assigned_Reviewer_1

This paper proposed a latent Baeysian melding approach which uses an individual model to incorporate population information with latent variable. The authors thoroughly presented the mathematical modeling of the method and applied the proposed approach to two electricity datasets and showed they could reach lower error compared with the base line method.

The paper is solid in math and the models and parameter estimation are presented in detail. The results clearly show consistent reduction in modelling error both on synthetic and real data. The significance and other field of application of the work could be more clearly discussed by the authors. I put a few days to read through the math and they seem correct to me. To be frank, I am not quite sure about the significance of the work.

In summary the paper is well written and the derivations and models seems correct. The results are promising. Though it would be interesting to have more discussion of the other applications of the new modeling and its audience.
Summary: The paper is solid in math and the models and parameter estimation are presented in detail.

Submitted by Assigned_Reviewer_2

This paper proposes a method to integrate two prior belief under different views and applies it to a specific engineering problem called energy disaggregation. The method is based on an approach called Bayesian melding proposed previously. The authors replace a fixed prior in Bayesian melding with a prior having latent variables, and MAP-estimate the latent variables. The result shows the proposed method may improve the separation of energy consumption timeseries of individual appliances from total energy.

The energy disaggregation problem is interesting in its own right and it is understandable that the integration of different types of prior belief should be important for this kind of problem. However, the proposed method is seen just an application of Bayesian melding once the latent variables in the prior have been fixed, so the method is quite trivial from general point of view. On the other hand, the method might contribute to the specific field where the energy disaggregation problem is relevant, while it was unclear to me from the manuscript.

Although the paper is well written overall, the presentation of the background and the explanation of Bayesian melding (Sections 1 and 2) should be improved. The introduction is not very specific so it was difficult at the first reading to see what those methods mentioned here (posterior regularization, learning with measurements, etc. and also Bayesian melding) will actually do. Section 2 is quite problematic as it only gives the formula of Bayesian melding. The authors should introduce when and why having two different priors should be reasonable, and how it is related to the notion of individual and population mentioned in Introduction. Some more justification about the form of \tilde{p}_S(S) should also be necessary.

Minor comments: - Sec.2: usuall -> usually, invertable -> invertible - Give the definition of c_\alpha - Is there any reasonable method to select alpha?
Summary: The problem to be solved is interesting but the method is a trivial extension of existing one. The contribution is not very clear.

Submitted by Assigned_Reviewer_3

This paper proposed latent bayesian melding (LBM) approach by using the Bayesian melding idea, which is tailored to a specific problem of energy disaggregation (using AFHMM). Connections between LBM and posterior regularization (PR) are made, which is also interesting. Results demonstrated the benefits of LBM over baselines (AFHMM with/without PR).

Quality: A couple of comments- a. When using the melding idea to construct the prior using \tau=f(S), f is not invertible in general models and problems. It is suggested in the paper we use Monte Carlo method to get S_1..N and use Gaussian distribution to approximate p(\tau). How much complexity does this add (if the generative model is more sophisticated, i.e., f() is harder to compute)? What N did you use in the experiment, does it affect much? b. \alpha seems a trade-off between the two sources of priors; how sensitive is the result to \alpha, i.e., if you vary \alpha from 0 to 1, how much variation did you observe? How did you choose this in practice?

Clarity: The paper is well written; method and experiment setup are clearly stated.

Originality: Introducing Bayesian melding approach into machine learning is fairly new. The AFHMM (with and without PR) for energy disaggregation problem is standard, but ok to be used as a demonstration of the method.

Significance: If other examples or models can be provided to support the general LBM approach, the work would be even more signifiant.
Summary: This work brings the Bayesian melding idea into attention and makes the connection with posterior regularization, which provides some insight on constructing priors for real-world problems. The use case (energy disaggregation problem) is clearly analyzed and fair comparison is made with a couple of baselines. I recommend an accept.

Author Feedback
Author rebuttal: Thanks to all reviewers for the very helpful feedback.

Assigned_Reviewer_2:
The reviewer is correct that a simple alternative to our approach would be to run MAP on the latent variables, and then hold the latent variables S fixed and Bayesian melding method on the model variables when fixing the latent variables. However, this is computationally expensive and does not scale to high dimensions, as the previous Bayesian melding method requires performing density estimation for the distribution tau. Instead we propose an approximate joint prior in Section 3, which allows us to infer the latent variables and model parameters jointly. Thus our algorithm scales better than the original Bayesian melding algorithm.

Why it is interesting to have two different distributions on the same quantity: When modelling the same phenomenon at different levels of generality, analysts tend to use different kinds of models. For example, consider the application of energy disaggregation. At the individual level, we can get readings of the household's power usage at two-minute intervals, which suggests a time series model. But at a higher level, if we want to model how a household's energy usage changes as a function of the type of dwelling, socio-economic status, etc., it makes more sense to regress these variables on the daily or monthly usage (e.g., by a generalized linear model). The problem tackled into our paper is a procedure that combines these two types of models in a way that is computationally efficient and crucially does not require much effort on the part of the human analyst.

Assigned_Reviewer_3:
In our experiment we set N=1000. All the functions f() are linear which required the least computation. We could make a comparison when varying N in the future. For the \alpha, we set them equal for those priors. The inference for \alpha is very difficult in general. We did not focus on this in this paper. This could be an interesting topic for future research.

Assigned_Reviewer_9:
Our hope is that the significance will be that there are many other potential applications in which one would want to use separate models for populations and for individuals (for example, Bayesian melding has been applied to genetics), and that this algorithm would also be useful in those settings.